# Probiotics as a Friendly Antibiotic Alternative: Assessment of Their Effects on the Health and Productive Performance of Poultry

**Rafiq Ahmad** [1,†]**, Yu-Hsiang Yu** [1,†]**, Felix Shih-Hsiang Hsiao** [1]**, Andrzej Dybus** [2]**, Ilyas Ali** [3]**, Hui-Chen Hsu** [1,*]
**and Yeong-Hsiang Cheng** [1,*]

1   Department of Biotechnology and Animal Sciences, National Ilan University, Yilan 26047, Taiwan
2   Department of Genetics, West Pomeranian University of Technology, 70-311 Szczecin, Poland
3   Department of Medical Cell Biology and Genetics, Health Sciences Center, Shenzhen 518060, China
*   Correspondence: hchsu@niu.edu.tw (H.-C.H.); yhcheng@niu.edu.tw (Y.-H.C.)
†   These authors contributed equally to this work.

**Abstract:** Antibiotics have been used to maintain the overall health of poultry by increasing production efficiency, promoting growth, and improving intestinal function for more than 50 years. However, they have a number of side effects, such as antibiotic resistance, gut dysbiosis, destruction of beneficial bacteria, and the potential to spread diseases to humans. In order to address the aforementioned issues, a lot of effort is put into the development of antibiotic alternatives. One of them is the use of probiotics that can be added to the feed in order to increase poultry performance and avoid the aforementioned problems. Probiotics are live microorganisms consumed as feed additives or supplements. They function in the poultry gastrointestinal tract to benefit the host. Probiotics improve growth performance, bone health, meat and eggshell quality. The addition of probiotics to the diet also positively affects the immune response, intestinal microflora, and disease resistance. Careful selection of probiotic strains is of utmost importance. This review focuses on the significance of probiotics as a potential antibiotic-free alternative and the way in which they can be used as supplements in poultry feed for boosting production and safeguarding health.

**Keywords:** intestinal morphology; reproduction; antibiotic resistance; probiotics; poultry

## 1. Introduction

Over the past seventy years, the improvement of feed consumption and genetic selection have been the primary areas of poultry research [1]. The control of a variety of microbial infectious diseases caused by *Streptococcus*, *Staphylococcus*, *Escherichia coli*, *Pseudomonas*, *Campylobacter*, *Salmonella*, *Yersinia*, *Bacillus*, *Clostridium*, *Mycobacterium*, *Enterococcus*, *Klebsiella* and *Proteus* species has been less thoroughly investigated [2]. The immune system of broilers is not fully developed during the first few weeks and therefore it is more susceptible to bacterial infection [3]. Furthermore, it can take up to eight weeks for the gut microbiota to develop and stabilize. The longer the time necessary to reach bacterial homeostasis, the greater the risk of bacterial infection [1]. Poultry are kept in closed facilities to minimize the risk of bacterial infection [4]. In many cases, farmers continue to supplement feed with antibiotics [5]. In the past decades, in traditional commercial poultry production, antibiotics have been one of the most frequently used additives to improve feed conversion, growth rate, and bird health, thereby increasing profitability and productivity [6,7]. Poultry constitute the largest global population of food animals [8]. Antibiotics kill susceptible bacteria (microbes) in any poultry system, leaving behind some resistance genes that can be passed on to other bacteria [9]. Antibiotic resistance results from the ability of these resistant bacteria to spread from one host to another, either directly or indirectly [10]. Antibiotics exert negative effects on human and animal welfare when used excessively

at subtherapeutic levels or continuously for an extended period of time [11]. Antibiotic resistance in bacteria can develop in a variety of ways, including a decreased permeability of bacterial cell membranes, changes in the antibiotic binding sites and enzyme production. It can also be acquired from other bacterial species present in the environment [12]. Numerous bacteria have developed antibiotic resistance as a result of inappropriate antibiotic use [13]. Oxacillin and tetracycline resistance has been found in *Staphylococci* directly linked to poultry farms. Some species of *Staphylococcus* commonly infecting poultry and causing staphylococcosis, pododermatitis, and septicemia have developed resistance to β-lactam antibiotics [14]. *Pseudomonas aeruginosa* strains isolated from Ghanaian poultry were resistant to cephalosporins, carbapenems, penicillin, quinolones, monobactam, and aminoglycosides [15]. In Nigeria, *Pseudomonas aeruginosa* showed resistance to β-lactams, tetracycline, nitrofurantoin, tobramycin, and sulfamethoxazole-trimethoprim [16]. In a similar way, ciprofloxacin, erythromycin, ceftriaxone, meropenem, and colistin resistance was found in Pakistani poultry [17]. Furthermore, *Escherichia coli* has increased its resistance to the majority of poultry-specific medications, such as tetracycline [18]. Different *Salmonella* species have been found to be resistant to ampicillin, tetracycline, trimethoprim, ciprofloxacin, and sulfamethazole [19]. In the same way, it has been reported that *Campylobacter jejuni* and *Escherichia coli* were resistant to erythromycin and tetracycline [1,20]. On the other hand, the involvement of antibiotic growth promoters in the emergence of multi-drug-resistant microorganisms has raised concerns for global public health. The development of antibiotic-resistant microbial populations in animal populations led to the potential transfer of antibiotic-resistant genes from animals to humans [21]. Therefore, many European countries forbade the use of antibiotics in poultry feed in 2006 [22]. Similarly, the US Food and Drug Administration issued Veterinary Feed Directives in 2015. They recommended the limited application of antibiotics only for animal treatment [23]. In Sweden, antimicrobial medications for growth promotion and prophylaxis were banned in 1986 and 1988, respectively [1]. South Korea was the first country in Asia to forbid the use of antibiotic growth promoters in animal feed in July 2011 [22]. The restriction of antibiotic use in feed raises the demand for alternatives to avoid a sharp decline in animal productivity and economic losses. In the past two decades, probiotics, prebiotics, synbiotics, phytobiotics, enzymes, essential oils, and fatty acids have gained widespread popularity among poultry nutritionists. Among them, probiotics have been shown to improve immune function, gut morphology and physiology. This, in turn, increases poultry performance and well-being. Feed supplements known as probiotics contain live beneficial bacteria like *Bifidobacterium*, *Lactobacillus*, and *Streptococci*, yeast such as *Candida* and *Saccharomyces*, and fungi like *Aspergillus awamori*, *A. niger*, and *A. oryza*, all of which have the potential to maintain the balance of intestinal microflora, and stimulate the immune system [24]. Some well-known probiotics of bacterial origin include *Bifidobacterium*, *Lactobacillus*, *Streptococci*, and *Bacillus subtilis*, which show antimicrobial activity against some pathogenic species, such as *Escherichia coli*, *Clostridium perfringens*, *Staphylococcus aureus*, and *Salmonella typhimurium* [25]. This review describes the ways in which probiotics boost growth and health and discusses the benefits of including them as feed supplements in poultry diet.

## 2. Probiotics and Growth Performance

The pathogen most prevalent in the intestines of chickens, particularly broilers, is *Salmonella*. Therefore, probiotics have been potential candidates for growth promoters in the majority of commercial poultry diets since the withdrawal of antibiotic growth enhancers in poultry nutrition. Antibiotic growth promoters act by blocking the production and secretion of pro-inflammatory cytokine-degrading intermediates in the gastrointestinal tract, resulting in the disturbed gut microbiota [26]. Probiotics, on the other hand, alter the gut environment and strengthen its barrier function through the immune system stimulation, as presented in Table 1 [27]. A total of 280 females of Japanese quails were fed with a mixture of without rapeseed meal, non-fermented post-extraction rapeseed meal (5%, 10%, 15%), and a fermented one (5%, 10%, 15%). The data analysis revealed that the

addition of 10% fermented rapeseed meal had the most beneficial effects as such egg quality traits as egg weight, specific gravity, yolk index and color, and albumen pH [28]. In broilers, probiotic non-pathogenic bacteria compete with the pathogenic ones for nutrients in the gut. They also colonize the intestines, preventing pathogenic bacteria from inhabitation and stimulating the secretion of digestive enzymes (e.g., β-galactosidase, α amylase), thus facilitating the absorption of nutrients, and enhancing broiler growth performance [29]. An increased average dietary feed consumption and conversion result in an improved body weight gain, which in turn affects production performance, even though probiotics do not always significantly influence feed consumption and feed conversion. All of the above-mentioned processes depend on several factors, including strain selection and application, time concentration, as well as the absorption of dietary probiotics [30]. An increased average dietary feed consumption and an enhanced feed conversion efficiency are closely attributed to an improvement in body weight gain, which in turn complements production performance [31]. The application of dietary supplementation of probiotics improves body weight gain and feed conversion, even though probiotics do not always significantly enhance feed consumption [32]. The body weight gain, average daily diet consumption, feed conversion efficiency, and production performance of the poultry birds are influenced by several potential factors including strains selection, application, time concentration, and absorption of dietary supplementation of probiotics [33].

**Table 1.** Summary of the beneficial probiotics on poultry performance.

| Probiotic Strains | Biological Performance | Reference |
|---|---|---|
| *Bacillus amyloliquefaciencs* | Improve intestinal health and growth performance | [34,35] |
| *Bacillus coagulans* | Enhances growth performance and gut health | [36] |
| *Lactobacillus acidophillus* | Improve production performance and helps the immune system and gut histomorphology | [37,38] |
| *Lactobacillus bulgaricus* | Enhances growth performances and improves immune functions | [39] |
| *Pediococcus acidilactici* | Improves laying performances and modulates intestinal microflora composition | [40,41] |
| *Propionibacterium acidipropionic* | Contributes to the better development of intestinal mucosa and microbiota composition | [42] |
| *Saccharomyces cerevisiae* | Improves growth performance and enhances laying performance | [43] |
| *Streptococcus faecium* | Avoided the impairment and regulated the stability of the epithelial intestine, and improves the immune functions | [44] |

Wang et al. [45] immunized hatched chicks with a strain of *Lactiplantibacillus plantarum LT-113*, and found that it protected against *Salmonella typhimurium* by limiting gastrointestinal invasion and inhibiting tight junction gene expression in intestinal cells. In the control group, *Salmonella* infection compromised the intestinal mucosal barrier. On the other hand, Olnood et al. [46] revealed that the oral administration of *Lactobacillus johnsonii* decreased *Salmonella* and *Clostridium perfringens* invasion in the gastrointestinal system. Additionally, it has been demonstrated that the combination of xylanase and a multistrain probiotic enhanced dietary energy absorption in the intestine and its preservation in the liver [47]. Energy changes may result from improved nutrient digestibility and feed conversion rate. Probiotics increased synthesis of short-chain fatty acids, stimulated the immune system and metabolism [48,49].

Short-chain fatty acid (SCFA) metabolites produced during microbial carbohydrate fermentation in the gastrointestinal tract impact leukocytes and endothelial cells by stim-

ulating G-protein-coupled receptors and inhibiting histone deacetylase. SCFAs increase the level of IgA produced by B immune cells, impede the NF-κB transcription factors and reduce the production of proinflammatory cytokines [50,51]. Dietary supplementation of poultry with *Bacillus licheniformis*, a facultative anaerobic bacterium, enhances the gastrointestinal tract absorption rate and surface area. It also stimulates the growth and multiplication of probiotic bacteria such as *Lactobacillus*, *Bifidobacterium*, and *Aspergillus awamori*, as shown in Figure 1 [52].

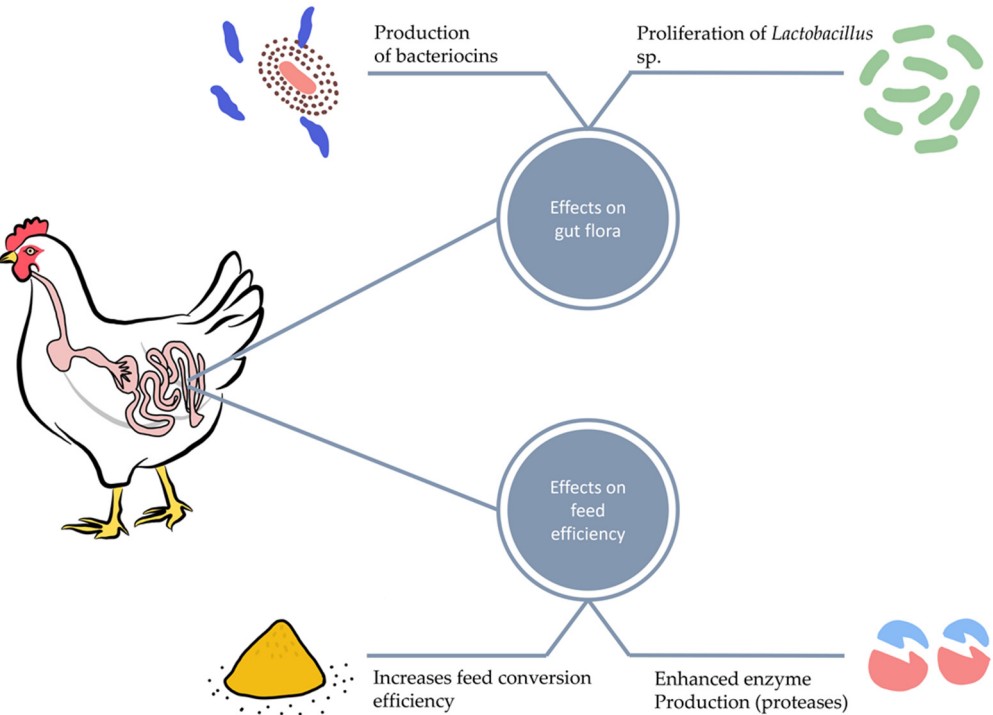

**Figure 1.** Effect of probiotics on growth performance, gut macroflora and feed efficiency.

## 3. Probiotics and Intestinal Morphology

Poultry health status and improved growth efficiency are strongly correlated with the gut condition and intestinal microflora. The intestines play a very important role in the digestive tract of birds as they harbor a diverse community of beneficial microbes, which degrade complex nutrient compounds into simpler molecules that are more easily assimilated and metabolized [53,54]. The structural organization and adherence properties of the gastrointestinal epithelial cells are crucial for nutrient absorption and the protection of the bird's body from pathogenic microbes that could infiltrate the bloodstream [55]. The most important parameters associated with the higher nutritional absorption resulting from larger surface area available for nutrient assimilation are those related to intestinal morphology, i.e., increased villus height, a lower crypt depth, and the higher villus height to crypt depth ratio [56]. Sound gastrointestinal microbiota is the prime requirement for avoiding microorganism infections in the gastrointestinal tract of birds. This is achieved by preventing microorganism colonization through pathogenic bacteria antagonism, inhibition of adhesion sites in the gut and impediment to bacterial exercises [57]. Similarly, another marker of gastrointestinal health status is the amount of gastric mucosa, which produces mucin and prevents pathogenic organisms from adhering to mucosal surfaces [58]. Even though the intestinal microflora is relatively stable, it is still affected by numerous environmental factors (feed composition, hygienic standards, physical stress, etc.) and overall health condition of the animal. However, the key element with the greatest impact on intestinal microflora is diet. Probiotics are commonly used for intestinal flora regulation [59] and improvement of gastrointestinal histomorphology; however, the potential effects may slightly differ from one strain to another [60]. Dietary supplementation of broilers with

*Lactobacillus plantarum* and *Lactobacillus reuteri* significantly affected barrier activity and reduced colonization by certain opportunistic or pathogenic microorganisms [61].

Zheng et al. [36] exposed broilers to *Salmonella enteritidis* (SE) and found a significant decline in goblet cell membranes at 7 days post-infection (DPI), as well as a reduction in villus height and villus-crypt ratio in the small intestine. In contrast, birds fed dietary supplementation of *Bacillus coagulans* had a relatively low crypt depth, a greater villus-crypt ratio, and a larger number of goblet cells in the jejunum at 7 and 17 days post-infection. Gastrointestinal mucous cells synthesize mucin-2, a constituent of mucus, which facilitates the enhancement of barrier activity in *Salmonella enteritidis*-infected birds. Supplementation with a *Bacillus licheniformis*-fermented product at 1.25 and 5 g/kg improved cecal morphology and increased the survival rate of broilers and conserve a stable number of goblet cells in the ileum as well as in the caecum under *Eimeria tenella* challenge. A 1.25 g/kg dose reduced lesions scores in the cecum, while that of 5 g/kg decreased the oocyst-count index. Furthermore, surfactin C isolated from *Bacillus licheniformis*-fermented products inhibited *Eimeria* oocyst sporulation and disrupted sporozoite morphology [62].

## 4. Probiotics and Immune Response

The chicken requires a strong immune system for optimal performance. The immune system comprises lymphoid organs located in the different parts of the body. In addition to highly specialized lymphoid structures like Meckel's diverticulum, the bursa of Fabricius, cecal tonsils, and Peyer's patches, which are connected to the gastrointestinal lumen, numerous lymphoid cells can be found in the epithelial mucous membrane (intraepithelial lymphocytes) [63,64]. Enteric neurons and gut immune cells communicate with each other in order to coordinate their actions against stressors [65]. Among the neuroendocrine compounds produced by the hypothalamic-pituitary-adrenal and sympathetic-adrenal medullary axes are corticosterone and catecholamines, which have the potential to influence immune regulation and phagocyte activity in a variety of immune cells [55]. Physical restrictions such as a low gastric pH and rapid transit in the small intestine play a crucial role in preventing pathogens from colonizing the gastrointestinal tract and causing inflammation [58]. In addition, pathogens must overcome the physical barriers imposed by the epithelium and intestinal microflora as well as the response of the host defense system in order to ultimately cause infection [66]. Cristofori et al. [67] showed that some non-pathogenic gut microbiota altered physiological cellular responses and the ability of an organism to fight infections by interacting with the host defense system and epithelium. Other potential advantages of probiotics are based on their significant impact on the intestinal environment. Epithelial and dendritic cells that constitute sentinel cells in the mucosa are found in lymphoid tissue connected to the intestinal tract. The binding of probiotic microbe-associated molecular patterns to Toll-like receptors on sentinel cells triggers NF-kB and MAP kinase pathways [67]. This activation does not only exert cytoprotective effects but also increases or inhibits the expression of genes controlling the inflammatory process by stimulating, signaling, and interpreting antimicrobial factors [68]. Additional advantages include enhanced epithelial barrier function, bacterial adhesion to the intestinal epithelium, and inhibition of microbial adhesion [58]. Probiotics are considered potential alternatives to antibiotics for improving immune health and growth performance in broilers. Cheng et al. [69] reported that dietary supplementation with *Bacillus licheniformis* enhanced T-cell immunity without impairing bird growth. It also directly impacted chemokine expression of genes and enhanced the production of pro- and anti-inflammatory cytokines in the mucosal surface, which had a profound effect on the immune system. Probiotics also influence immune function by affecting B-lymphocytes. Two bioactive secondary metabolites produced by probiotic bacteria, short-chain fatty acids, and bacteriocins, prevent infectious agents from growing and surviving [70]. Notably, several *Lactobacillus* strains producing lactic acid were found to be able to lower the pH level of their surroundings. Lie et al. [34] observed that *Bacillus amyloliquefaciens* initially reduced the stress caused by the immune response in lipopolysaccharide-challenged broiler chickens and increased

plasma lysozyme activity and WBC count in 192-day-old males. Consequently, *Bacillus amyloliquefaciens* was able to restore impaired immune status and growth performance [71]. Yitbarek et al. [72] fed a mix of probiotics obtained from various strains of *Bacillus subtilis*, *Lactobacillus acidophilus*, and *Lactobacillus casei* and prebiotics (yeast-derived carbohydrates) to 300-day-old Lohman pullets. In this case, synbiotics enhanced the immune system and maintained homeostasis through an IL-10-specific response. Synbiotic supplementation resulted in the increased concentrations of IL-6, interferon-γ (IFN), and IL-4 in the ileum.

Hetab et al. [73] demonstrated that the production of antibodies against the Newcastle disease virus in layers was significantly enhanced by probiotic bacteria (*Bacillus subtilis* and *Enterococcus faecium*). Therefore, broiler chickens supplemented with *B. subtilis* showed higher levels of antibodies against Newcastle disease, infectious bronchitis, and bursal disease [74].

## 5. Mode of Action Probiotics

### 5.1. Probiotics and Competitive Exclusion

Bacteria attempt to eradicate pathogens harmful to the gastrointestinal tract due to their natural competitiveness, which is called bacterial intervention, competitive exclusion (CE), or bacterial belligerence [71]. Probiotics, prebiotics, and synbiotics exhibit the property of competitive exclusion. The Nurmi concept, which later evolved into the competitive exclusion concept, is based on the introduction of gastrointestinal flora into young chickens in order to induce bacterial resistance to pathogenic microbes [75]. These microbes colonize cell attachment sites or mucosal surfaces and disrupt microbiome composition in the gut causing intestinal infections. On the other hand, probiotics are capable of adhering to the inner mucosal layer, greatly increasing the amount of time during which they may remain in the gastrointestinal tract [2,55]. Therefore, probiotic bacteria occupy more space in this tract, thus eliminating pathogens through competition. Consequently, birds are able to consume more nutrients. Competitive exclusion is usually considered to occur in the caeca and intestines of birds [76]. Supplementation with *Bacillus amyloliquefaciens* (BAP) for 35 days (20 g/kg) significantly improved the growth of broiler chickens due to the facilitated food digestion, nutrient absorption and availability in a healthy digestive system [77]. A form of competitive exclusion, i.e., the oral administration of spores, primarily from the genus *Bacillus*, may support and enhance host defense against infectious diseases. The potential molecular biological mechanisms of probiotic action and competitive pathogen elimination are presented in Figure 2 [44].

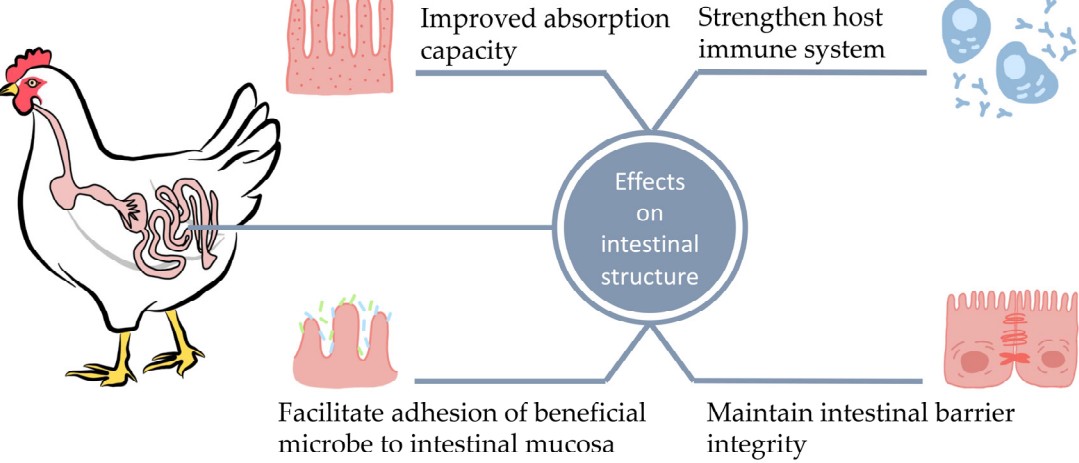

**Figure 2.** Effects of probiotics on intestinal health and immune function.

### 5.2. Probiotics and Organic Acid (Acidity and pH)

Organic acids or acidifiers are naturally occurring compounds with acidic properties that could be defined as weak carboxylic acids (R-COOH) such as acetic, propionic, lactic,

formic, fumaric, and sorbic acids. It has been stated that the inclusion of organic acids improved growth performance, feed efficiency, and the digestibility of nutrients [78]. Additionally, organic acids may play a role in both suppressing the colonization of pathogenic bacteria and triggering the immune system [79]. These enhancements could be made through lowering the pH of the GIT, increasing the utilization of nutrients in diets, suppressing the growth and proliferation of pathogens, and increasing the immune responsiveness of poultry [80].

From the scientific evidence, it is hypothesized that probiotics and organic acids may work together to enhance beneficial bacteria in the GIT and protect against pathogenic bacteria. Previously it has been reported that feeding broiler chickens a diet high in organic acids and probiotics had a number of promising effects on growth performance, energy and protein utilization, and gut microflora [81]. It has the ability to penetrate the cellular cytoplasm. In addition to inhibiting bacterial cell enzymes such as decarboxylase and catalases, the acid disintegrates within the cell cytoplasm [82]. In order to increase the production and distribution of organic acids (such as lactic acid and acetic acids) in the intestinal tract of monogastric animals, bacterial probiotics such as propionic acid and fumaric acid can be added to their diet, lowering the pH of the gastrointestinal tract. This may improve the intestinal microbial environment for some native microorganisms and mitigate the invasion of pathogenic microbes [83]. Furthermore, probiotic strains are capable of competitive exclusion, thus preventing the evolution of pathogenic bacteria.

Some probiotic strains have a remarkable capacity to endure hostile environments in their hosts. They are able to pass through the digestive tract and survive under extremely acidic conditions, such as gastric acid and bile [84]. This is quite difficult since the pH in the stomach of many animals ranges from 1.5 to 3.0. Furthermore, bile salts and several digestive and intestinal enzymes significantly contribute to the disruption of microbiota balance [85]. Nevertheless, it has been demonstrated that spores can sprout properly and survive all the way through the gastrointestinal tract being attached to feed particles, which play a protective role. Re-sporulation is the most common way for bacteria to stay alive during their transit in the animal body and diet seems to influence spore sprouting and propagation, since it contains the nutrients necessary for their survival [86]. Propionic acid and fumaric acid supplementation lowers the pH of the local gut environment and prevents the growth of certain pathogens like *Escherichia coli*, *Salmonella typhimurium*, and *Clostridium perfringens*, enhancing nutrient absorption and immunity, ultimately leading to improved physical and productive performance in different poultry species [87].

### 5.3. Probiotics and Gut Microbiota

Probiotics have a significant impact on the composition and function of the gut microbiome by competing with other microbes for nutrient content, binding to sites and receptors on the intestinal mucosa, and producing antimicrobial agents to inhibit the growth of other microbes [71]. Potential mechanisms for the antagonistic activity of probiotics include the lowering of gut pH, modulation of the immune system, and production of organic acid [51]. Probiotics can also significantly improve intestinal barrier integrity, maintain immunogenicity and affect microbial signaling pathways in intestinal epithelial cells [88].

A variety of tools are used to investigate the effects of probiotics on gut microbiota function, variation, and composition, including culture-dependent methods, metagenomic sequencing, and in vivo assays. However, in vivo probiotic administration is the most effective and efficient technique for obtaining precise results [89]. Several studies have shown that probiotics, especially lactic acid bacteria, can effectively prevent *Salmonella Enteritidis* and *Escherichia coli* 078:K80 infections in poultry [90].

Furthermore, the enforcement of the diet with probiotics has been reported to strengthen the composition of the gut microbiota by limiting pathogen proliferation and increasing the number of beneficial microorganisms. Abdel-Moneim et al. [91] found that in ovo inoculation with *Bacillus bifidum*, *Bacillus animalis*, *Bifidobacterium longum*, and *Bifidobacterium infantis* improved ileal bacterial composition by enhancing the intestinal colonization

with bacterial species such as *Lactobacillus* and *Bifidobacterium*, while decreasing the total amount of *coliforms* bacteria. Abou-Kassam et al. [92] also observed that the potential addition of *Bacillus toyonesis* and *Bacillus bifidum* to the diet impeded the growth of fungi and *coliforms* and decreased the number of *E. coli* and *coliforms* in the caecum, whereas Anas Abdelqader et al. [93] reported that dietary supplementation with *Bacillus subtilis* reduced the abundance of *Clostridium* and cecal *coliforms*, while preventing the multiplication of *coliforms* in the diet.

### 5.4. Application and Validation of Probiotics Secondary Metabolites

Probiotics are live microorganisms that benefit the health of their hosts when applied in sufficient quantities [94]. Numerous reports have addressed their use in poultry farming and human health. A number of microbes, particularly fungi and bacteria, possess probiotic properties; however, *Bifidobacterium*, *Lactobacillus*, *Streptococcus*, and *Bacillus* sp. are the most frequently used ones as presented in Table 2 [95]. The significance of *Bacillus*-derived probiotics to industry depends on several factors, including their high safety level, quick growth rate, short fermentation time, and capacity to secrete proteins into extracellular medium [96]. Different *Bacillus* species, such as *Bacillus licheniformis*, *Bacillus subtilis*, *Bacillus cereus*, and *Bacillus claussi,* are utilized as probiotics in poultry diets.

**Table 2.** An overview of the use of bacteria beneficial in poultry farming.

| Probiotic | Biological Performance | Reference |
|---|---|---|
| *Lactiplantibacillus plantrum* LTC-113 | Enhances immunity against *Salmonella typhimurium*, and preserves intestinal epithelial barrier function | [45,97] |
| *Lactobacillus johnsonii* | Alleviates *Salmonella sofia* and *Clostridium perfringens* infection | [46,98] |
| *Bacillus subtilis* C-3102 | Decreases the number of *Salmonella enterica* serovars (*enteritidis* LM-7) | [99,100] |
| *Pediococcus acidilactici* | Decreases the number of *Salmonella enterica* serovars (*Gallinarum*) | [101,102] |
| *Lactobacillus acidophilus* | Improves body weight, decreases mortality, enhances the immune response in *Escherichia coli* 0157-challenged chickens | [103,104] |
| *Bacillus subtilis* | Increases the ratio of villus height to crypt depth, surface area available for nutrient absorption in the duodenum and ileum, the number of *Blautia*, *Faecalibacterium* and *Romboutsia*, and the amount of beneficial microflora (*Lactobacillus*, *Bifidobacterium*, and *Enterococcus*) | [105,106] |
| *Bacillus subtilis* PB-6 | Boosts plasma calcium and phosphorus concentrations, broiler production and welfare, increases bone mass and meat quality | [93,107] |
| *Bacillus subtilis* DSM29784 | Increases the number of *Butyricicoccus* and *Faecalibacterium* in the intestine, improves health, weight, and the tight junction complex in broilers suffering from necrotic enteritis | [108,109] |

*Bacillus*-derived peptides have been demonstrated to possess antifungal, antimicrobial, anticarcinogenic, antiviral, anti-amoebic, and anti-mycoplasmic properties [110]. It has been shown that *Bacillus subtilis*, one of the most significant aerobic bacteria, exerts positive effects on poultry diets by limiting the spread of aerobic pathogens and increasing the efficiency of diet protein. Extracellular digestive enzymes produced by *Bacillus subtilis* may

enhance the immune response and function and the development of the gastrointestinal tract [111], increasing internal egg quality and decreasing the cholesterol content of egg yolks [112].

Cheng et al. [113] reported that 4 days of *Bacillus subtilis*-fermented products containing the highest concentration of surfactin showed the greatest antimicrobial activity against pathogens like *Clostridium perfringens*, *Staphylococcus aureus*, *Escherichia coli*, and *Salmonella typhimurium*. In broilers infected with *Clostridium perfringens*, dietary supplementation with *Bacillus subtilis*-fermented products containing surfactin significantly affected gastrointestinal tract morphology and healed ulcerated lesions. *Bacillus subtilis* treatment could boost broiler growth and productivity. It also improved bone quality, gastrointestinal structure, and function. Cheng et al. [114] found that surfactin isolated from *Bacillus subtilis*-fermented products was a prospective antibiotic and antibacterial agent substitute, which exerted significant antibacterial effects against *Brachyspira hyodysenteriae* by altering its morphological characteristics and preventing bacterial growth. Additionally, it reached maximum activity against *Clostridium perfringens*. Surfactin derived from the fermented products of *Bacillus licheniformis* inhibited in vitro growth of *Clostridium perfringens* in a dose-dependent manner. Broilers challenged with the above-mentioned bacterium showed significant improvements in body weight and average daily weight gain when supplemented with *Bacillus licheniformis*-fermented products (2 g/kg). They also benefited from reduced necrotic lesions and improved intestinal tract morphology [69]. Moreover, surfactin derived from *Bacillus licheniformis*-fermented products was more effective against *Clostridium perfringens* than that obtained from the *Bacillus subtilis*-fermented products.

In a similar way, *Lactobacillus*-based probiotics increased the number of goblet cells in the duodenum and jejunum of broilers. By decreasing goblet cell proliferation and differentiation and regulating mucin mRNA expression, probiotics are said to increase the number of goblet cells [115]. Broilers fed probiotics containing *Lactobacillus casei*, *Lactobacillus acidophilus*, *Bifidobacterium thermophilum* and *Enterococcus faecium* had increased villus height and a lower crypt depth [44].

Additionally, dietary supplementation of broilers with *Bacillus coagulans* [36], *Lactobacillus reuteri* and *Lactobacillus salivarius*, propionic bacterium *acidopropionic* [42], mixture of *Bacillus licheniformis*, *Bacillus subtilis*, *Saccharomyces cerevisiae* [116] and *Pediococcus acidilactici* [117] improved villus length and the ratio of villus height to crypt depth, suggesting that probiotics may increase nutrient absorption. On the other hand, acetic and benzoic acids lowered lesion scores in broiler chickens challenged with the different *Eimeria* species [118,119]. The former had also anticoccidial properties against *Eimeria tenella*. Oocysts were adversely impacted by lowering the pH levels of the caeca, ultimately resulting in lower lesion scores [119].

One of the organic acids, lactic acid, produced by the bacteria fermenting feed carbohydrates lowers the pH of the local environment and prevents the growth of certain pathogens like *Escherichia coli*, *Salmonella typhimurium*, and *Clostridium perfringens* [87]. Finally, blends of acetic, butyric, and lactic acids increased feed conversion ratio and weight gain in 7-day-old broilers exposed to *Clostridium perfringens* [120], whereas *Lactobacillus johnsonii*-based probiotics improved intestinal development and microbiota balance in birds challenged with *Clostridium perfringens* [121].

In broiler chicken, dietary supplementation of *Enterococcus faecium PNC01* improved ileal villus height and crypt depth and decreased the comparative length of the cecum at day 21 and enhanced the relative length of the jejunum and ileum at day 42. Additionally, *Enterococcus faecium*-supplemented diets enhanced the relative abundance of *Firmicutes* and *Lactobacillus* and decreased the relative abundance of Bacteroides in the cecal microflora [122]. Svetoch et al. [123] conducted a research study on bacteriocin producer *Enterococcus faecium* that peptide class IIa bacteriocin E50-52 have minimal inhibitory concentration against *Clostridium jejuni*, *Yersinia* specie, *Salmonella* specie, and *Escherichia coli* ranged from 0.025 to 32 μg/mL. In the therapeutic broilers trail, oral supplementation with E50-52 decreased *Clostridium jejuni* as well as *Salmonella enteritidis* by more than

100,000 times in the caeca, and systemic *Salmonella enteritidis* was reduced in both liver and spleen.

Volzing et al. [124] demonstrated that recombinant *Lactococcus lactis* that produce and secreted heterogenous antimicrobial peptides A3APO and Alyteserin showed maximum inhibitory activity against pathogenic bacteria *Escherichia coli* and *Salmonella*. Volzing et al. observed that A3APO and Alyteserin containing recombinant *Lactococcus lactis* inhibited the growth of the pathogenic bacteria *Escherichia coli* and *Salmonella* by up to 20-fold while sustaining the host's viability. Yang et al. [125] documented that, immunized chickens with recombinant invasive *Lactobacillus plantarum* against coccidiosis induced greater levels of specific antibodies in the serum and the secretory IgA (SIgA) was increased in the intestinal washes. Furthermore, a higher proportion of $CD^{4+}$ and $CD^{8+}$ T cells were detected in the peripheral blood. These results demonstrate that recombinant *Lactobacillus palantarum* effectively activated immune responses against *E. tenella* infection. Therefore, it should be emphasized that probiotic supplements aid in combating pathogens, enhancing nutrient absorption and immunity, ultimately leading to improved physical and productive performance in different poultry species.

## 6. Conclusions

Probiotics improve gut health by promoting the activity of digestive enzymes, which increase nutrient digestibility and growth performance in poultry. This suggests that they can be used as growth promoters. Additionally, probiotics protect the host from pathogens by regulating immunomodulatory response and utilizing a competitive exclusion strategy to prevent the colonization of the gastrointestinal tract by pathogenic microorganisms. However, the full elucidation of probiotic effects at a molecular level and the interactions between probiotics, pathogens, and epithelial cells needs further investigation. This review points to the involvement of metagenomic, metabolomic, and proteomic research and analysis in determining the biological effect of probiotics. Therefore, the revelation of previously unknown facts will enable the thorough understanding of the role of probiotics in the growth and health of poultry.

**Author Contributions:** Conceptualization, Y.-H.C.; methodology, R.A., Y.-H.Y., F.S.-H.H., A.D. and H.-C.H.; data curation, R.A. and Y.-H.Y.; writing—original draft preparation, R.A. and Y.-H.Y.; writing—review and editing, I.A. and Y.-H.C.; visualization, R.A. and Y.-H.Y.; supervision, Y.-H.C.; project administration, R.A., Y.-H.Y. and Y.-H.C. All authors have read and agreed to the published version of the manuscript.

**Funding:** This research was funded by Ministry of Science and Technology, Taiwan, grant number 108-2321-B-197-001.

**Data Availability Statement:** Not applicable.

**Acknowledgments:** We are grateful to Daniel Zabroski, Department of Ruminant Science, West Pomeranian University of Technology, Poland, who polish the English of the manuscript.

**Conflicts of Interest:** The authors declare no conflict of interest.

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
