# Peer review of "Probiotics as a Friendly Antibiotic Alternative: Assessment of Their Effects on the Health and Productive Performance of Poultry"

_fermentation, doi:10.3390/fermentation8120672_

Round 1

Reviewer 1 Report

This review deals with an interesting topic, present an extensive bibliography, and illustrates numerous aspects concerning this subject.

However, the paper requires a thorough revision both in terms of language and content.

The paper is often approximate (for example no reference is ever made to figure 1) repetitive and confusing.

This work cannot be accepted in this form.

Author Response

Revision Notes

These revision notes are disclosed to address comments/reviews on "fermentation-2023192", entitled: Probiotics as a friendly antibiotic alternative: assessment of their effects on the health and productive performance of poultry

Introduction:

The authors would like to express their gratitude to MDPI Fermentation for permitting us to revise our manuscript, thereby improving the quality of the revised manuscript before the high standard of this publication channel.

The authors attempted as best as possible to strike balance between conciseness and results reporting. The aim is to create an informative paper for the readers that clearly articulate the key components. The remarkable improvements of the revised manuscript are:

Major and comprehensive clarifications have been made based on the comments from the Reviewers, as well as they have been implemented in the corresponding section of the revised manuscript.

The modified/revised parts of the manuscript are

Reviewer 1 Comments:

Comment 1: Comments and Suggestions for Authors

This review deals with an interesting topic present an extensive bibliography and illustrate numerous aspects concerning this subject.

However, the paper requires a thorough revision both in terms of language and content.

The paper is often approximate (for example no reference is ever made to figure 1) repetitive and confusing.

This work cannot be accepted in this form.

Response to Comment 1:

The authors would like to thank the reviewer for the careful and thorough reading of this manuscript and for the thoughtful comments and constructive suggestions, which help to improve the quality of this manuscript. We have thoroughly checked and improved the technicality of the manuscripts and incorporated the possible suggested comments by Reviewer 1. According to the Reviewers’ suggestion, changes have been made to the revised version of the manuscript.

The article has been revised both in terms of language and content.

Figure 1 reference has been given and cited in the text in lines 143-144.  

Reviewer 2 Report

This article is written in correct English and has interesting summary elements, but it needs to be restructured to gain readability and quality.

My comments in ascending order of importance.....

First : There are sometimes overlaps between paragraphs.  For example in the chapter 3 line 221"According to studies, Bacillus licheniformis conserve a stable number of goblet cells in the ileum as well as in the caecum [55]" the sentence would be better placed in the next chapter entitled "4. Probiotic and Intestinal morphology".

Second,  If the article mentions a lot of bibliography for the modes of action of probiotics, some articles are still missing on the effect of probiotics on production performance in chapter3 (especially lines 190-199). Thus a summary table  on the influence of the main probiotics on poultry performance is missing (equivalent to Table 1 on mode of action) and would make the article much better. 

Third point: the  majority of references are about bacilli, the article should rebalance  with more references on lactic acid bacteria or enterrococcus faecium for example. In a generic article on probiotics, why talk about secondary metabolites of bacillus, and not antimicrobial peptides of lactic acid bacteria? Or restrict the field of the article to bacillus only as they are significant part of the market at the moment.

Fourth: the Application and validation of Bacillus secondary metabolites section needs to be completely reviewed. It is confusing, poorly structured, it talks about pigs (lines 372-378), and lactic acid bacteria (lines 384-389) which are irrelevant. Or please change the scope of the paragraph.

While the first point can be considered minor, the other three are major and require significant editing of the article.

Author Response

Revision Notes

These revision notes are disclosed to address comments/reviews on "fermentation-2023192", entitled: Probiotics as a friendly antibiotic alternative: assessment of their effects on the health and productive performance of poultry”

Introduction:

The authors would like to express their gratitude to MDPI Fermentation for permitting us to revise our manuscript, thereby improving the quality of the revised manuscript before the high standard of this publication channel.

The authors attempted as best as possible to strike balance between conciseness and results reporting. The aim is to create an informative paper for the readers that clearly articulate the key components. The remarkable improvements of the revised manuscript are:

Major and comprehensive clarifications have been made based on the comments from the Reviewers, as well as they have been implemented in the corresponding section of the revised manuscript.

The modified/revised parts of the manuscript are

Reviewer 2 Comments (Highlighted with green text)

Comment 1: Comments and Suggestions for Authors

Comments and Suggestions for Authors

This article is written in correct English and has interesting summary elements, but it needs to be restructured to gain readability and quality.

Response to Comment 1:

The authors would like to thank the reviewer for the careful and thorough reading of this manuscript and for the thoughtful comments and constructive suggestions, which help to improve the quality of this manuscript. We have thoroughly checked and improved the technicality of the manuscripts and incorporated the possible suggested comments by Reviewer 2. According to the Reviewers suggestion following changes have been made in the revised version of the manuscript.

The article has been revised both in terms of language and content.

Comment 2:

First: There are sometimes overlaps between paragraphs.  For example, in chapter 3 line 221"According to studies, Bacillus licheniformis conserve a stable number of goblet cells in the ileum as well as in the caecum [55]" the sentence would be better placed in the next chapter entitled "4. Probiotic and Intestinal morphology".

Response to Comment 2: Updated

Comment 3:

Second, If the article mentions a lot of bibliography for the modes of action of probiotics, some articles are still missing on the effect of probiotics on production performance in the chapter 3 (especially lines 190-199). Thus, a summary table on the influence of the main probiotics on poultry performance is missing (equivalent to Table 1 on the mode of action) and would make the article much better. 

Response to Comment 3:

Updated

Comment 4:

Third point: the majority of references are about bacilli, the article should rebalance with more references on lactic acid bacteria or enterococcus faecium for example. In a generic article on probiotics, why talk about secondary metabolites of bacillus, and not antimicrobial peptides of lactic acid bacteria? Or restrict the field of the article to bacillus only as they are significant part of the market at the moment.

Response to Comment 4:

Updated

Comment 5:

Fourth: The Application and validation of Bacillus secondary metabolites section needs to be completely reviewed. It is confusing, poorly structured, it talks about pigs (lines 372-378), and lactic acid bacteria (lines 384-389) which are irrelevant. Or please change the scope of the paragraph.

Response to Comment 5:

Updated

While the first point can be considered minor, the other three are major and require significant editing of the article.

Reviewer 3 Report

The manuscript “Probiotics as a friendly antibiotic alternative: critical appraisal for health and productive performance of poultry“ is interesting.

It may be published after a few corrections / additions.

Comments

Please consider changing the title to

"Probiotics as a friendly antibiotic alternative: assessment of their effects on the health and productive performance of poultry"

Line 40-42 - I do not agree with this statement. Can't be generalized. Since antibiotics have not been used for feed in EU countries since 2006, the exception is medicated feed. - please edit it. The authors write about it later in the paper, but I think that it should be mentioned at the beginning.

Line 127-128 - please change it - explain - I don't know what the authors meant.

Line 136-137 - The authors write, " Organic acid supplementation resulted in decreased lipid peroxidation in Eimeria tenella-challenged broilers …” lipid peroxidation reduction but what? - cell membrane? - please specify it.

Line 136-137. Please also specify which organic acids - this is too general a statement.

In chapter 2.2, mention should be made of the mechanism by which probiotics stimulate the formation of organic acids.

The authors do not cite Figure 1 and Table 1 in the text.

Chapter 2.3. - I miss an attempt to explain the role of organic acids produced by probiotic bacteria, e.g. in digestion / absorption of nutrients.

Line 263 - please specify what fermented products are.

Line 268 - what was the medium on which the bacillus from which sulfactin was isolated was grown? - please explain it in the text - the comment also applies to chapter 6

Line 320 and 588 - please verify the author's name.

Line 322-323 - does synbiotic weaken the immune system? - please verify it

Please check the literature carefully.

Recently, a publication has appeared on the effectiveness of fermented products in feeding quails. I believe that its content would be a valuable supplement to the information contained in the manuscript.

Wengerska K, Czech A, Knaga S, Drabik K, Próchniak T, Bagrowski R, Gryta A, Batkowska J. The Quality of Eggs Derived from Japanese Quail Fed with the Fermented and Non-Fermented Rapeseed Meal. Foods. 2022 Aug 18;11(16):2492. doi: 10.3390/foods11162492. PMID: 36010492; PMCID: PMC9407498.

Other Notes:

Due to the fact that this article is addressed to the journal "Fermentation", I believe that the authors should include more information in the work linking probiotics with fermented products, which have recently been incorporated into poultry nutrition with great interest.

The article is interesting, consistent and can be published after a few corrections and additions.

Author Response

Revision Notes

These revision notes are disclosed to address comments/reviews on "fermentation-2023192", entitled: Probiotics as a friendly antibiotic alternative: assessment of their effects on the health and productive performance of poultry

Introduction:

The authors would like to express their gratitude to MDPI Fermentation for permitting us to revise our manuscript, thereby improving the quality of the revised manuscript before the high standard of this publication channel.

The authors attempted as best as possible to strike balance between conciseness and results reporting. The aim is to create an informative paper for the readers that clearly articulate the key components. The remarkable improvements of the revised manuscript are:

Major and comprehensive clarifications have been made based on the comments from the Reviewers, as well as they have been implemented in the corresponding section of the revised manuscript.

The modified/revised parts of the manuscript are

Reviewer 3 Comments (Highlighted with red text)

Comment 1: Comments and Suggestions for Authors

The manuscript “Probiotics as a friendly antibiotic alternative: critical appraisal for health and productive performance of poultry” is interesting.

It may be published after a few corrections/additions.

Please consider changing the title to

"Probiotics as a friendly antibiotic alternative: assessment of their effects on the health and productive performance of poultry"

Response to Comment 1:

The authors would like to thank the reviewer for the careful and thorough reading of this manuscript and for the thoughtful comments and constructive suggestions, which help to improve the quality of this manuscript. We have thoroughly checked and improved the technicality of the manuscripts and incorporated the possible suggested comments by Reviewer 3. According to the Reviewers suggestion following changes have been made in the revised version of the manuscript.

Updated: “Probiotics as a friendly antibiotic alternative: assessment of their effects on the health and productive performance of poultry"

Comment 2:

Line 40-42 - I do not agree with this statement. Can't be generalized. Since antibiotics have not been used for feed in EU countries since 2006, the exception is medicated feed. - please edit it. The authors write about it later in the paper, but I think that it should be mentioned at the beginning.

Response to Comment 2:

Corrected: In past decades

Comment 3:

Line 127-128 - please change it - explain - I don't know what the authors meant.

Response to Comment 3:

Corrected:

Comment 4:

Line 136-137 - The authors write, " Organic acid supplementation resulted in decreased lipid peroxidation in Eimeria tenella-challenged broilers …” lipid peroxidation reduction but what? - cell membrane? - please specify it.

Response to Comment 4:

Corrected:

Comment 5:

Line 136-137. Please also specify which organic acids - this is too general a statement.

Response to Comment 5:

Corrected:

Comment 6:

In chapter 2.2, mention should be made of the mechanism by which probiotics stimulate the formation of organic acids.

Response to comment 6: Propionic acid and fumaric acid

Comment 7:

The authors do not cite Figure 1 and Table 1 in the text.

Response to comment 7: Corrected: Figure 1 have been cited in the text in line 143-144 and Table 1 in line 100.

Comment 8:

Chapter 2.3. - I miss an attempt to explain the role of organic acids produced by probiotic bacteria, e.g. in digestion/absorption of nutrients.

Response to comment 8:

Corrected

Comment 9:

Line 263 - Please specify what fermented products are.

Response to comment 9: Bacillus licheniformis-fermented products

Comment 10:

Line 268 - what was the medium on which the bacillus from which surfactin was isolated was grown? - please explain it in the text - the comment also applies to chapter 6

Response to comment 10: Surfactin C was isolated from Bacillus licheniformis-fermented product. Chapter 6 corrected.

Comment 11:

Line 320 and 588 - please verify the author's name.

Response to comment 11:

Corrected: Yitbarek

Comment 12:

Line 322-323 - does synbiotic weaken the immune system? - please verify it

Response to comment 12:

Corrected

Comment 13:

Recently, a publication has appeared on the effectiveness of fermented products in feeding quails. I believe that its content would be a valuable supplement to the information contained in the manuscript.

Wengerska K, Czech A, Knaga S, Drabik K, Próchniak T, Bagrowski R, Gryta A, Batkowska J. The Quality of Eggs Derived from Japanese Quail Fed with the Fermented and Non-Fermented Rapeseed Meal. Foods. 2022 Aug 18;11(16):2492. doi: 10.3390/foods11162492. PMID: 36010492; PMCID: PMC9407498.

Other Notes:

Due to the fact that this article is addressed to the journal "Fermentation", I believe that the authors should include more information in the work linking probiotics with fermented products, which have recently been incorporated into poultry nutrition with great interest.

The article is interesting, and consistent and can be published after a few corrections and additions.

Response to comment 13: Thank you so much for your valuable and precious comments and suggestions. We have incorporated valuable supplements and information from the aforementioned paper, and we have already mentioned and included probiotics with fermented product information, but we have reused the probiotics with fermented products in terms of fermented products throughout the article, especially in chapter 5.4

Round 2

Reviewer 1 Report

This paper is much improved and clearer

Reviewer 2 Report

Many improvements and clarifications. Thanks